Development and validation of the AUDEXCEL algorithm as a diagnostic tool for occupational noise-related hearing disorder

Yew Sheng Qian ysq_sam@yahoo.com
Raja Pathmanathan Pothanantha
Department of Public Health Medicine, Faculty of Medicine, Universiti Kebangsaan Malaysia , Cheras, Kuala Lumpur , Malaysia
Marunaka Yoshinori
Electronic publication date: 2025 Oct 10
Publication date: 2025
Volume: 13
Electronic Location ID: e20149
Received 2025 May 8; Accepted 2025 Sep 5
Copyright: © 2025 Yew and Raja Pathmanathan
Copyright year: 2025
Copyright holder: Yew and Raja Pathmanathan
License: This is an open access article distributed under the terms of the Creative Commons Attribution License, which permits unrestricted use, distribution, reproduction and adaptation in any medium and for any purpose provided that it is properly attributed. For attribution, the original author(s), title, publication source (PeerJ) and either DOI or URL of the article must be cited.
License URL: https://creativecommons.org/licenses/by/4.0/

Keywords: Hearing, Occupational, Worker, Algorithm, Diagnostic

Funding: National University of Malaysia FF-2024-461 This work was supported by the National University of Malaysia Fundamental Grant (FF-2024-461). The funders had no role in study design, data collection and analysis, decision to publish, or preparation of the manuscript.

==============================
Background

Occupational noise-related hearing disorder (ONRHD) is an occupational disease that poses a significant challenge for workers both globally and locally. To address this, the study aimed to develop and validate AUDEXCEL, an Excel-based algorithm designed to diagnose ONRHD among workers in Malaysia.

Materials & Methods

This cross-sectional study involved audiograms from 320 workers. These audiograms were first analyzed by experienced occupational health doctors (OHDs) to establish a gold standard diagnosis. The same audiograms were then assessed using the AUDEXCEL algorithm, which was validated through 5-fold cross-validation. Validity (sensitivity, specificity, positive predictive value, and negative predictive value) and reliability (Cohen’s kappa coefficient) were evaluated for normal hearing, hearing loss, hearing impairment, permanent standard threshold shift (PSTS), temporary standard threshold shift (TSTS), and noise-induced hearing loss (NIHL).

Results

The sensitivity, specificity, positive predictive value (PPV), and negative predictive value (NPV) for normal hearing, hearing loss, PSTS, and TSTS were all 100.0%, with perfect agreement (κ = 1.000). For hearing impairment, sensitivity, specificity, PPV, and NPV were 100.0%, 99.5%, 99.5%, and 100.0%, respectively, with almost perfect agreement (κ = 0.994). For NIHL, sensitivity, specificity, PPV, and NPV were 90.3%, 98.0%, 95.0%, and 96.1%, respectively, also showing almost perfect agreement (κ = 0.922).

Conclusions

AUDEXCEL demonstrated high validity and reliability in replicating expert diagnoses and may serve as a supportive diagnostic aid for ONRHD. However, its use should be complemented by clinical and occupational exposure assessments due to the inherent complexity of diagnosing ONRHD.

Background

Occupational noise-related hearing disorders (ONRHD) is defined as partial or complete hearing loss resulting from one’s employment (Centers for Disease Control and Prevention (CDC), 2025). It is an umbrella terminology that consists of hearing loss (i.e., a partial or complete inability to perceive sound in which a pure tone threshold >25 dB at any audiometric frequency), hearing impairment (i.e., a permanent shift of ≥25 dB in the pure tone audiometry), permanent standard threshold shift (PSTS), temporary standard threshold shift (TSTS), and noise-induced hearing loss (NIHL) (Department of Occupational Safety and Health (DOSH) of Malaysia, 2019). Globally, the prevalence of ONRHD is notably high, with prevalence ranging from 11.2% to 58.0% (Chen, Su & Chen, 2020).

On top of its high prevalence, ONRHD is found to exert adverse impacts on workers’ productivity and work safety. Communication difficulties stemming from partial or complete hearing loss in one or both ears impede comprehension, particularly in noisy environments. This will impede interactions with colleagues and supervisors and potentially disrupting effective teamwork and job coordination (Metidieri et al., 2013). Concerning work safety, affected workers may encounter challenges in promptly responding to auditory warnings or signals, thereby compromising industrial safety and overall job performance. Not to mention, this will increase the risk of occupational accidents (International Labour Organization (ILO), 2024). Additionally, ONRHD may lead to reduced situational awareness, especially in settings with heavy machinery, elevating the risk of occupational injuries (International Labour Organization (ILO), 2024). On top of these, non-auditory effects of ONRHD such as perceived annoyance, cognitive decline, sleep disruptions, and impacts on cardiovascular health (Basner) may reduce the work productivity and increase sickness presenteeism.

From an economic perspective, the Global Burden of Disease 2010 highlighted that hearing disorders, in general, affect around 1.3 billion people worldwide and represent the 13th highest contributor to years lived with disability (YLD), accounting for 19.9 million YLDs, or 2.6% of the global burden (Vos et al., 2012). By 2020, hearing disorders in the general population resulted to an estimated loss of 62,218 Quality-Adjusted Life Years (QALYs) and 135,561 Productivity-Adjusted Life Years (PALYs), which translate to an economic impact of approximately USD 3.6 billion loss due to diminished well-being and USD 13.9 billion loss due to reduced productivity (Si et al., 2020). Regarding compensation costs, a single ONRHD claim amounted to USD 6,438, which generally covered medical expenses such as diagnostic evaluations, hearing aids, rehabilitation, hearing aid fittings, batteries, accessories, and follow-up care (Makaruse, Maslin & Shai Campbell, 2025).

Due to the high prevalence as well as the adverse work and economic impacts of ONRHD, conducting noise risk assessments is a mandatory practice across all industries to identify potential excessive noise exposure. If a workplace is found to exceed permissible noise levels, employers are obligated to provide annual audiometric testing for their workers (Department of Occupational Safety and Health (DOSH) of Malaysia, 2019). These tests are typically carried out at occupational health clinics or through mobile audiometric testing center (ATC). The resulting audiograms are then reviewed and interpreted manually by occupational health doctors (OHDs), who are responsible for diagnosing ONRHD based on article-based records.

Although the conventional paper-based approach to diagnose ONRHD has been utilised for an extended period, it comes with several limitations, particularly in cases involving numerous parties and workers in the workplace. Firstly, it is time-consuming, particularly for OHDs who manage large companies with numerous workers, such as automobile manufacturing plant (Colsman et al., 2020). This can lead to delays in notifying regulatory bodies and employers. Secondly, interpreting audiometric test results requires expertise, and junior OHDs may struggle with accuracy, potentially resulting in misdiagnoses (Walker et al., 2013). Thirdly, the OHDs often have difficulty deciding whether a particular audiogram needs to further action, such as to be repeated or reported to the authority (Department of Occupational Safety and Health (DOSH) of Malaysia, 2019).

Given the above limitations, there is a need for an automated and yet simple algorithm to streamline and automate diagnosis of ONRHD (Wasmann et al., 2022). Its core functionality is to facilitate ONRHD diagnosis, ensure compliance with occupational health and safety regulations, mitigate ONRHDs, and enhance overall workers’ well-being. The algorithm must be able to assist OHDs in diagnosing ONRHD within a large number of workers within a short period of time. Therefore, this study aimed to develop the AUDEXCEL algorithm for diagnosing ONRHD among workers in Malaysia. In addition, it sought to evaluate the validity and reliability of the AUDEXCEL algorithm in identifying ONRHD in this population.

Despite the abovementioned objectives, it should be highlighted that diagnosing ONRHD, particularly NIHL, is inherently complex due to several factors. Firstly, standard pure tone audiometry may not capture supra-threshold auditory deficits such as cochlear synaptopathy (Kamerer et al., 2019). Secondly, there is significant inter-individual variability in susceptibility to noise-induced damage, influenced by genetic, metabolic, and environmental factors (Chen et al., 2022). Thirdly, age-related hearing loss (also known as presbycusis) often overlaps with NIHL, especially in older workers, making it difficult to distinguish the primary etiology based on audiometric patterns alone (Fetoni et al., 2022). Given these challenges, ONRHD diagnosis typically requires a combination of audiometric evaluation, occupational noise exposure history, and exclusion of other etiologies. While this study focuses on audiogram-based pattern recognition, it is crucial to understand that such an approach represents only part of a broader diagnostic process.

Materials and Methods

A cross-sectional study was conducted from January 2025 to March 2025.

Study population

Audiograms of the workers, who attended an occupational health clinic in Kuala Lumpur, Malaysia were obtained. To be eligible for inclusion in the present study, the workers must adhere to the following criteria: (i) Aged between 18 and 60 years old; (ii) workers of any gender or ethnic background; and (iii) workers must have baseline and annual measurements. However, workers with the following characteristics will be excluded for participation: (i) Workers experiencing cognitive impairment; (ii) workers who underwent ear surgeries at the time of recording; and (iii) workers with a history of notable ear disorders (such as Meniere’s disease or acoustic neuroma) that could impact hearing; and (iv) workers with presbycusis. The inclusion and exclusion criteria of these workers were ascertained via the review of their medical history (such as history of presenting illness, underlying medical history, social history, and occupational history) by the researchers.

Sample size estimation

The present study developed an excel-sheet based ONRHD diagnostic tool, named as the AUDEXCEL algorithm, which then undergo validity and reliability tests. The validity of the AUDEXCEL algorithm in diagnosing ONRHD was assessed through a concurrent validity test, via the sensitivity and specificity tests. As per Bujang & Adnan (2016), a diagnostic study typically aims for both sensitivity and specificity to be predetermined at a minimum of 70.0%. With an ONRHD prevalence of 51.4% (Yap et al., 2023), a precision of 0.05, a confidence interval of 95%, and expected sensitivity and specificity of 70%, the required sample size was calculated to be 320.

Meanwhile, the reliability of the AUDEXCEL algorithm in diagnosing ONRHD was assessed through inter-rater reliability, utilising Cohen’s Kappa coefficient. With a minimum acceptable Cohen’s kappa of 0.4 and an expected Cohen’s kappa of 0.6 (McHugh, 2012), along with an ONRHD prevalence of 51.4%, a power of 80%, and a significance level of 0.05, the required sample size was determined to be 156 using an online sample size calculator (Arifin, 2025). Since the sample sizes for validity and reliability are 320 and 156, respectively, the larger sample size (i.e., 320) was selected for this study.

Sampling technique

A random sampling method was employed, where 320 audiograms (considered as the sampling unit) were randomly selected from the occupational health clinic’s database, using an online random number generator.

Development of the AUDEXCEL algorithm

AUDEXCEL was created in the form of an Excel spreadsheet containing six distinct algorithms. Hearing impairment is characterised by a permanent shift of 25 dB or more in the pure tone average (PTA) of hearing thresholds at 500, 1,000, 2,000, and 3,000 Hz, relative to the standard reference level of 0 dB (Department of Occupational Safety and Health (DOSH) of Malaysia, 2019). Accordingly, the first algorithm, which aims to detect hearing impairment, was developed using the following formula:

PTA=Thresholdat500Hz+1,000Hz+2,000Hz+3,000Hz4.

According to World Health Organization (WHO), hearing loss is defined as a partial or complete inability to perceive sound, indicated by a pure tone threshold (PTT) greater than 20 dB at any tested audiometric frequency (World Health Organization (WHO), 2021). Specifically, mild hearing loss falls within the range of 21 to 34 dB, moderate hearing loss falls between 35 and 49 dB, moderately severe hearing loss ranging from 50 to 64 dB, severe hearing loss characterised by thresholds between 65 and 79 dB, profound hearing loss falls between 80 to 94 dB HL, and complete or total hearing loss falls above 95 dB. As a simplification (i.e., to classify an audiogram as “yes” or “no” for hearing loss), the present study defines hearing loss as a pure tone threshold (PTT) greater than 20 dB at any tested audiometric frequency. Hence, the second algorithm, designed to detect hearing loss, was therefore developed using the following formula:

PTT(500Hzor1,000Hzor2,000Hzor3,000Hzor4,000Hzor6,000Hzor8,000Hz)>20dB.

A standard threshold shift (STS) refers to a shift of 10 dB or more in the pure tone average (PTA) at 2,000, 3,000, and 4,000 Hz when compared to the baseline audiogram (Department of Occupational Safety and Health (DOSH) of Malaysia, 2019). This shift can be either temporary or permanent. An STS is considered permanent if the threshold shift remains present in a follow-up audiogram conducted within 3 months of the initial test. Based on this definition, the third algorithm (for PSTS) and the fourth algorithm (for TSTS) were developed using the following formula:

PTA=Thresholdat2,000Hz+3,000Hz+9,000Hz3.

Although various diagnostic criteria exist for NIHL, the present study adopted the criteria proposed by Hoffman et al. (2006) and McBride & Williams (2001). According to these definitions, NIHL is identified when the following conditions are met: (i) Threshold worse by ≥15 dB at 3,000, 4,000 or 6,000 Hz than the average thresholds at 500 and 1,000 Hz; (ii) V-shaped notch of ≥15 dB occurring at one audiometric frequency and (iii) ≥10 dB recovery at the high frequency. These criteria were translated into the following formula for the fifth algorithm:

PTT=(3,000Hz−1,000Hz≥15dB)and(4,000Hz−500Hz≥15dB)and(4,000Hz−1,000Hz≥15dB)and(6,000Hz−500Hz≥15dB)and(6,000Hz−1,000Hz≥15dB)and(8,000Hz−3,000Hz≥10dB)and(8,000Hz−4,000Hz≥10dB)and(8,000Hz−6,000Hz≥10dB).

Finally, the sixth algorithm for normal hearing is defined using the following formula:

PTT= (NotHearingLoss)and(NotHearingImpairment)and(NotPSTS)and (NotTSTS)and(NotNIHL).

The AUDEXCEL interface, shown in Fig. 1, enables OHDs to input audiogram data for hundreds or even thousands of workers simultaneously, delivering immediate diagnoses for ONRHD. To determine PSTS and TSTS, both baseline and annual audiograms must be available for each worker. AUDEXCEL generates separate diagnostic results for the left and right ears. It also alerts OHDs if a specific audiogram needs further actions, such as to be repeated or reported to the relevant authority. Furthermore, the tool features a real-time graphical display of the audiogram, enhancing visual interpretation and ease of analysis.

Figure 1 (A–C) Interface of the AUDEXCEL algorithm.

Validity and reliability tests of AUDEXCEL algorithm

In assessing the validity and reliability of the AUDEXCEL algorithm in diagnosing ONRHD, the diagnoses made by OHDs served as the benchmark or gold standard. All 320 audiograms were first analysed by 10 experienced OHDs to determine the diagnoses, whereby each OHD will receive 32 anonymised audiograms. These OHDs were recruited from a local medical university and the Ministry of Health, Malaysia. OHDs who fulfil the following criteria were recruited: (i) Actively practice occupational health service; (ii) qualified as occupational health doctor for at least 5 years; (iii) willing to provide consent for the currently study. Each of these 320 audiograms corresponds to a single worker and had at least one diagnosis. Some audiograms consisted of more than one diagnosis. Examples of audiograms and their diagnoses were presented in Table 1 below.

Table 1 Number of audiograms for each hearing diagnosis.

Audiograms	ONRHD diagnosed by OHDs	
Worker 1	Normal hearing	
Worker 2	Hearing loss and hearing impairment	
Worker 3	PSTS and NIHL	
Worker 4	TSTS and NIHL	
Worker 5	NIHL	

To evaluate the diagnostic performance of the AUDEXCEL algorithm, a 5-fold cross validation procedure was employed (Fig. 2). The dataset, comprising 320 audiograms, was randomly partitioned into five equal subsets (folds) of 64 audiograms each. In each iteration, four folds (256 audiograms) were used as training set for the algorithm, while the remaining fold (64 audiograms) served as the testing set. This process was repeated five times, with each fold used exactly once as the testing set.

Figure 2 A five-fold cross validation of the AUDEXCEL algorithm.

Statistical analyses

The algorithm’s performance was assessed during each iteration. The final performance metrics were obtained by averaging the results across all five folds, providing a more robust and unbiased estimate of AUDEXCEL’s generalisability to new data.

The validity of the AUDEXCEL was assessed using sensitivity, specificity, positive predictive values (PPV), negative predictive value (NPV) for each diagnosis category (normal hearing, hearing impairment, hearing loss, PSTS, TSTS, and NIHL) by comparing the diagnoses made by the AUDEXCEL algorithm with those made by the OHDs.

The reliability of the AUDEXCEL algorithm will be evaluated using an inter-rater reliability test, specifically Cohen’s kappa coefficient, κ. According to Landis & Koch (1977), Cohen’s Kappa values can be interpreted as follows: values of 0 or less suggest no agreement, 0.01 to 0.20 indicate slight agreement, 0.21 to 0.40 reflect fair agreement, 0.41 to 0.60 represent moderate agreement, 0.61 to 0.80 signify substantial agreement, and 0.81 to 1.00 denote almost perfect agreement (McHugh, 2012). In the present study, a kappa value above 0.4 (i.e., moderate agreement) was considered acceptable, although the researchers expected a kappa value above 0.6 (i.e., substantial agreement). These thresholds were selected to reflect a balance between methodological rigor and practical expectations in health-related studies involving subjective judgement. All statistical analyses were performed using SPSS version 28.

Ethical consideration

The National University of Malaysia Ethics Committee granted ethical approval to carry out the study within its facilities (Ethical Application Ref: JEP-2024-871). The researchers adhered to the principles outlined in the Declaration of Helsinki as well as the Malaysian Good Clinical Practice Guidelines.

The researchers notified the workers before using their audiograms, which were recorded in the occupational health clinic’s database. Online informed consent forms were signed by the workers before analysing their audiograms. In cases where the workers decline to consent, their audiograms were not utilised for analysis, and instead, they were replaced by audiograms from other consenting workers to reach the target sample size of 320 audiograms. Online information booklets outlining the study’s objectives and procedures as well as the informed consent forms were provided to the OHDs via working email.

Results

A total of 320 workers (audiograms) were employed to develop and validate the AUDEXCEL algorithm. The mean age of these workers was 41.7 ± 8.4 years old. A majority of them were male, Malaysian, and working in the machinery and equipment industries (Table 2).

Table 2 Sociodemographic characteristics of the workers who contributed their audiogram.

Sociodemographic characteristics	Mean ± SD	n (%) (N = 320)	
Age (years old)	41.7 ± 8.4	–	
Gender	
Male	–	227 (70.9)	
Female	–	93 (29.1)	
Nationality	
Malaysian	–	192 (60.0)	
Foreigner	–	128 (40.0)	
Working industries	
Machinery and equipment	–	133 (41.6)	
Textile, wearing apparel and leather	–	104 (32.5)	
Rubber	–	30 (9.4)	
Paper, printing and publishing	–	26 (8.1)	
Transport, vehicle and equipment	–	19 (5.9)	
Chemical	–	8 (2.5)	

Table 3 illustrates the diagnostic performance of AUDEXCEL in the training data. Of note, the sensitivity, specificity, PPV, and NPV of normal hearing were all 100.0%, with a perfect agreement (κ = 1.000). A similar trend was observed in hearing loss, PSTS, and TSTS. For hearing impairment, the sensitivity, specificity, PPV, and NPV were 99.9%, 99.6%, 99.5%, and 99.8%, respectively. The agreement for hearing impairment was almost perfect (κ = 0.994). For NIHL, the sensitivity, specificity, PPV, and NPV were 90.3%, 98.0%, 95.0%, and 96.1%, respectively. The agreement for NIHL was also almost perfect (κ = 0.891).

Table 3 Diagnostic performance of AUDEXCEL in the training data.

ONRHD	Sensitivity ± SD (%)	Specificity ± SD (%)	PPV ± SD (%)	NPV ± SD (%)	Cohen’s Kappa ± SD	
Normal hearing	100.0 ± 0.0	100.0 ± 0.0	100.0 ± 0.0	100.0 ± 0.0	1.000 ± 0.000*	
Hearing loss	100.0 ± 0.0	100.0 ± 0.0	100.0 ± 0.0	100.0 ± 0.0	1.000 ± 0.000*	
Hearing impairment	99.9 ± 0.3	99.6 ± 0.4	99.5 ± 0.4	99.8 ± 0.4	0.994 ± 0.003*	
PSTS	100.0 ± 0.0	100.0 ± 0.0	100.0 ± 0.0	100.0 ± 0.0	1.000 ± 0.000*	
TSTS	100.0 ± 0.0	100.0 ± 0.0	100.0 ± 0.0	100.0 ± 0.0	1.000 ± 0.000*	
NIHL	90.3 ± 9.0	98.0 ± 2.8	95.0 ± 6.9	96.1 ± 3.6	0.891 ± 0.021*	
Note:

* Statistically significant at p < 0.05.

Table 4 illustrates the diagnostic performance of AUDEXCEL in the testing data. The sensitivity, specificity, PPV, and NPV of normal hearing, hearing loss, PSTS, and TSTS, were all 100.0%, with a perfect agreement (κ = 1.000). For hearing impairment, the sensitivity, specificity, PPV, and NPV were 100.0%, 99.5%, 99.3%, and 100.0%, respectively. The agreement for hearing impairment was almost perfect (κ = 0.994). For NIHL, the sensitivity, specificity, PPV, and NPV were 89.5%, 100.0%, 100.0%, and 95.8%, respectively. The agreement was also almost perfect (κ = 0.922).

Table 4 Diagnostic performance of AUDEXCEL in the test data.

ONRHD	Sensitivity ± SD (%)	Specificity ± SD (%)	PPV ± SD (%)	NPV ± SD (%)	Cohen’s Kappa ± SD	
Normal hearing	100.0 ± 0.0	100.0 ± 0.0	100.0 ± 0.0	100.0 ± 0.0	1.000 ± 0.000*	
Hearing loss	100.0 ± 0.0	100.0 ± 0.0	100.0 ± 0.0	100.0 ± 0.0	1.000 ± 0.000*	
Hearing impairment	100.0 ± 0.0	99.5 ± 1.2	99.3 ± 1.7	100.0 ± 0.0	0.994 ± 0.014*	
PSTS	100.0 ± 0.0	100.0 ± 0.0	100.0 ± 0.0	100.0 ± 0.0	1.000 ± 0.000*	
TSTS	100.0 ± 0.0	100.0 ± 0.0	100.0 ± 0.0	100.0 ± 0.0	1.000 ± 0.000*	
NIHL	89.5 ± 6.5	100.0 ± 0.0	100.0 ± 0.0	95.8 ± 2.5	0.922 ± 0.048*	
Note:

* Statistically significant at p < 0.05.

Discussion

To the best of our knowledge, this is the first study to develop and validate a diagnostic tool of ONRHD in the form of an Excel sheet.

Notably, the diagnostic performance of AUDEXCEL, in terms of sensitivity, specificity, PPV, and NPV, ranging from almost perfect to perfect when identifying diagnoses of normal hearing, hearing loss, hearing impairment, PSTS, and TSTS. This high level of diagnostic accuracy can be attributed to the nature of AUDEXCEL as a rule-based and formula-driven algorithm, which directly applies established clinical criteria to audiometric data (Tokede et al., 2024). Unlike machine learning models, which rely on pattern recognition and probabilistic inference from large datasets (Sarker, 2021), AUDEXCEL uses explicit logical conditions grounded in well-defined audiological thresholds and classifications. As such, when the input audiograms are accurate and the clinical definitions are clear-cut, AUDEXCEL can produce diagnostic outputs that are consistently aligned with expert clinical judgment. This deterministic approach minimises variability and error, especially in datasets where the audiograms represent distinct and unambiguous diagnostic categories. Consequently, AUDEXCEL’s performance closely mirrors the gold standard provided by OHDs’ diagnoses, reflecting its potential reliability in structured clinical or occupational health settings.

However, such near-perfect diagnostic performance was not observed for the diagnosis of NIHL. This discrepancy may be explained by the fact that multiple diagnostic criteria exist for NIHL, and different OHDs may apply different standards in clinical practice. While AUDEXCEL primarily implements Hoffman et al. (2006) as well as McBride & Williams (2001) criteria, which is considered one of the most comprehensive and standardised frameworks for diagnosing NIHL, other OHDs might utilise alternative criteria such as those proposed by Coles, Lutman & Buffin (2000), Mahboubi et al. (2013) and Phillips, Henrich & Mace (2010). Each of these criteria incorporates slightly different thresholds and definitions for identifying NIHL on an audiogram, leading to variability in clinical judgment. For instance, Hoffman’s criteria defines NIHL as threshold worse by ≥15 dB at 3,000, 4,000 or 6,000 Hz than those at previous frequencies (Niskar et al., 2001), while Coles, Lutman & Buffin (2000) define NIHL as a threshold worse by ≥10 dB at 3,000, 4,000 or 6,000 Hz than those at previous frequencies. Such subtle but important differences in diagnostic thresholds mean that a single audiogram could be interpreted differently depending on which set of rules the clinician applies. Consequently, even though AUDEXCEL consistently applies Hoffman et al. (2006) as well as McBride & Williams (2001) criteria, discrepancies with the gold standard diagnosis made by different OHDs possibly using varied criteria.

To our knowledge, previous studies have applied machine learning models, such as support vector machines, decision trees, and neural networks, to detect or classify noise-induced hearing loss (NIHL) based on audiometric data (Soylemez et al., 2024). These approaches have demonstrated high accuracy and precision, especially when trained on large datasets with well-labeled outcomes (Soylemez et al., 2024). Comparatively, the current study adopts a different approach, which is not as advanced. However, the AUDEXCEL is a transparent, rule-based Excel algorithm, which mimics the decision-making logics based on established guidelines. Such logics can be easily reviewed, modified, or audited by professionals. Not to mention, they are accessible without the need for advanced computing skills or machine learning infrastructure, making it easily adapted by practitioners from lower- and middle-income countries (Yew et al., 2025).

Strengths and limitations

This study has several notable strengths. Firstly, it was based on a dataset of 320 real workers audiograms, which enhances the clinical relevance and internal validity of the AUDEXCEL algorithm. By working with real-world data, the findings are more likely to reflect actual clinical scenarios. Secondly, as abovementioned, AUDEXCEL employs a rule-based, formula-driven approach grounded in established clinical definitions. This method ensures transparency and reproducibility, distinguishing it from black-box models such as those used in machine learning. Furthermore, the algorithm achieved perfect diagnostic performance in terms of sensitivity, specificity, PPV, and NPV for identifying normal hearing, hearing loss, PSTS, and TSTS. This demonstrates its robustness and accuracy for these conditions. Another key strength is AUDEXCEL’s simplicity and accessibility. Unlike machine learning models that require extensive training and tuning, AUDEXCEL does not rely on large datasets or computational power. This makes it especially valuable in resource-limited settings or occupational health environments where ease of use and interpretability are critical. Additionally, the use of standardised diagnostic criteria, such as Hoffman et al. (2006) as well as McBride & Williams (2001) criteria for NIHL, provides a consistent basis for classification and facilitates comparison with published benchmarks.

Despite its strengths, the study also has important limitations. First of all, the diagnostic performance of AUDEXCEL was not as strong for NIHL, which may be due to variability in the criteria used by different OHDs. While AUDEXCEL follows the Hoffman et al. (2006) as well as McBride & Williams (2001) criteria, other OHDs may use alternative frameworks, which differ slightly in their diagnostic thresholds. As a result, discrepancies between AUDEXCEL and the clinical gold standard may reflect differing diagnostic philosophies rather than true misclassification. Secondly, since the same dataset was used both to develop and validate the algorithm, there is a risk of overfitting, which could lead to overly optimistic performance metrics. To mitigate overfitting within the dataset, the researchers employed a five-fold cross-validation approach, where the dataset was partitioned into five subsets. Each subset was used once as a validation set, while the remaining subsets were used for training the AUDEXCEL algorithm. Thirdly, the algorithm provides binary classifications (e.g., hearing loss vs. no hearing loss), which may not fully capture the spectrum or severity of hearing impairment as defined by the WHO. Another limitation is that while the AUDEXCEL algorithm was developed and validated based on audiogram patterns for diagnostic purposes, it does not directly incorporate the specific environmental noise exposure parameters such as duration, intensity (dB level), or types of noise at the workplace. The AUDEXCEL algorithm also does not account for supra-threshold auditory deficits, such as cochlear synaptopathy (Hockley et al., 2023).

Recommendations for future study

To enhance the utility and applicability of AUDEXCEL, several areas of improvement and future research are recommended for future studies. Firstly, future versions of AUDEXCEL could incorporate multiple diagnostic criteria for NIHL, allowing users to select or compare between criteria such Coles, Lutman & Buffin (2000). This flexibility would accommodate variations in clinical practice and improve agreement with diverse gold standards. Secondly, to address the risk of overfitting, it is recommended that AUDEXCEL be validated using an independent external dataset, ideally sourced from different clinical settings or populations. This would provide a more robust evaluation of its generalisability and performance in real-world us. Thirdly, future studies may investigate and expand the algorithm’s capability to classify degrees and types of hearing loss (e.g., mild, moderate, mixed, conductive vs. sensorineural), which would enhance its clinical value and align it more closely with audiological standards. Finally, the future development of AUDEXCEL could consider integrating workplace exposure parameters such as noise exposure duration, noise intensity, type of noise, and usage of personal hearing protection devices to improve its diagnostic utility and to support causal attribution more robustly.

Conclusions

AUDEXCEL showed excellent validity and reliability when compared to expert diagnoses in identifying various audiometric outcomes, including NIHL, hearing impairment, PSTS, and TSTS. While these findings suggest that AUDEXCEL may serve as a valid and reliable tool to support occupational health doctors in evaluating ONRHD, it should be applied in conjunction with clinical judgment and comprehensive noise exposure history. Future studies should further refine the algorithm by incorporating exposure metrics and testing in diverse populations.

Supplemental Information

Supplemental Information 1 Dataset consists of 320 audiograms from the workers.

Supplemental Information 2 AUDEXCEL Algorithm.

Supplemental Information 3 User Manual.

Supplemental Information 4 STROBE Checklist.

Additional Information and Declarations

Competing Interests

The authors declare that they have no competing interests.

Author Contributions

Sheng Qian Yew conceived and designed the experiments, performed the experiments, analyzed the data, prepared figures and/or tables, authored or reviewed drafts of the article, grant acquisition, and approved the final draft.

Pothanantha Raja Pathmanathan performed the experiments, analyzed the data, authored or reviewed drafts of the article, and approved the final draft.

Human Ethics

The following information was supplied relating to ethical approvals (i.e., approving body and any reference numbers):

The National University of Malaysia Ethics Committee granted ethical approval to carry out the study within its facilities (Ethical Application Ref: JEP-2024-871).

Data Availability

The following information was supplied regarding data availability:

The raw dataset, AUDEXCEL algorithm, and user manual of the AUDEXCEL are available in the Supplemental Files.

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
