# Peer review of "Development and validation of the AUDEXCEL algorithm as a diagnostic tool for occupational noise-related hearing disorder"

_PeerJ, doi:10.7717/peerj.20149_

## Round 0.1 · original submission · Major Revisions

·

Basic reporting

the idea of this paper is very good in developing an AUDEXCEL to facilitate the screening of workers at high and risky noise jobs.

no comments

Experimental design

no comments

Validity of the findings

no comments

Additional comments

as mentioned above, very good and new ideas for monitoring of noise induced hearing loss among different work areas, my main concern is the small numbers included in the study which needed to be generalized to examine the validity of the AUDECXEL.
second concern which is not mentioned in the study is the duration of noise the workers were exposed to it as well as the noise level among the work area and if this noise is steady or fluctuant.

·

Basic reporting

Line 43 and 43 - It will be highly rewarding if the authors could give operational definition of HL - Hearing Loss and HI - Hearing Impairment. Is there any difference? This will remove the ambiguity. Authors were able to define PSTS - Permanent Standard Threshold Shift, TSTS - Temporary Standard Threshold Shift and NIHL - Noise-Induced Hearing Loss.

Experimental design

Line 94 - Authors should clarify the inclusion criteria - (i) Aged between 18 and 60 years but in (ii) line 94 also - workers of any age. This needs further clarification. I believe workers (audiograms) who are older than 60 years were excluded because of the possibility of presbyacusis setting in, so, if it is then written that workers of any age, that is their audiograms should be included, it means there is a contradiction with the first (i) inclusion criterion.

Validity of the findings

No comment

Additional comments

Nil

Reviewer 3 ·

Basic reporting

I appreciate the authors’ effort to identify occupational hearing loss. While the manuscript is generally well-structured, several issues are apparent. The most significant concern is the authors' apparent lack of familiarity with audiological terminology, which undermines the scientific clarity of the work. This is the major shortcoming of the study. Specifically, the distinction—or lack thereof—between "hearing loss" and "hearing impairment" is unclear. These terms are often used interchangeably in the literature, and the authors should clarify what they precisely mean by each.

Additional comments are as follows:

Abstract:
The abbreviations PPV and NPV should be written out in full the first time they are mentioned.

Introduction:
The term noise-induced hearing loss (NIHL) is more appropriate than noise-related hearing disorders. Additionally, the authors should clarify the distinction between hearing loss and hearing impairment. NIHL is a sensorineural type of hearing loss, most commonly permanent in occupational contexts, and current treatments are rehabilitative (e.g., hearing aids). Therefore, NIHL impacts workers economically and functionally, and it also imposes a burden on employers and governments in terms of labor productivity and healthcare costs. The introduction would benefit from a brief discussion of such implications to highlight the importance of NIHL.

Methods:
The age range of 18–60 years is too broad for NIHL studies. The authors should explain how they accounted for or excluded presbycusis (age-related hearing loss). Moreover, a power analysis is necessary to demonstrate the adequacy of the sample size.

Discussion:
The discussion section is generally insufficient. Other machine learning studies that have attempted to predict NIHL should be addressed. For example:
Soylemez E, Avci I, Yildirim E, Karaboya E, Yilmaz N, Ertugrul S, Tokgoz-Yilmaz S. Predicting noise-induced hearing loss with machine learning: the influence of tinnitus as a predictive factor. J Laryngol Otol. 2024 Oct;138(10):1030–1035. doi:10.1017/S002221512400094X.

Overall comment:
While I find the manuscript potentially publishable, it is clear that the authors lack sufficient background in otolaryngology or audiological terminology. As a result, sections related to hearing loss are presented superficially, and some key terms and concepts are conflated or misused.

Experimental design

Research question well defined, relevant & meaningful. It is stated how research fills an identified knowledge gap.

Validity of the findings

All underlying data have been provided; they are robust, statistically sound, & controlled.

Additional comments

Overall comment:
While I find the manuscript potentially publishable, it is clear that the authors lack sufficient background in otolaryngology or audiological terminology. As a result, sections related to hearing loss are presented superficially, and some key terms and concepts are conflated or misused.

Reviewer 4 ·

Basic reporting

The abbreviation ONRHD is uncommon and may confuse readers. It is recommended to use the more widely accepted term NIHL (Noise-Induced Hearing Loss), or Occupational NIHL (ONIHL), which aligns with terminology used by authoritative bodies such as the World Health Organization (WHO) and the Centers for Disease Control and Prevention (CDC) CDC NIHL Prevention.

Several acronyms used throughout the manuscript are non-standard or self-invented, which reduces readability and may confuse readers. Standardized terminology should be used consistently.

The Introduction would benefit from the inclusion of more central and well-established references to better contextualize the study and align it with current literature. Suggestions could be: a) Vos T, et al. Years lived with disability (YLDs) for 1160 sequelae of 289 diseases and injuries 1990–2010: a systematic analysis for the Global Burden of Disease Study 2010. Lancet. 2012;380:2163–96. b) Basner M, et al. Auditory and non-auditory effects of noise on health. Lancet. 2014;383:1325–32. c) Lin FR. Hearing loss in older adults: who’s listening? JAMA. 2012;307:1147–8.

Experimental design

A critical issue that should be addressed early in the manuscript is the diagnostic complexity of ONIHL. Specifically, can ONIHL be reliably diagnosed based on the audiogram alone? Given the large inter-individual variability in susceptibility to noise exposure, and the fact that standard audiograms may not capture supra-threshold hearing deficits, this is a significant concern. The manuscript should introduce this diagnostic challenge in the Introduction, elaborate on it in the Discussion, and reflect it in the Conclusion. The current structure introduces this issue only in the Methods section (line 148), which is too late

• There is an inconsistency in the inclusion criteria: the manuscript states that participants must be aged 18–60 years (line 92), yet shortly thereafter refers to “workers of any age.” This contradiction should be clarified.
• The assessment of cognitive abilities is mentioned but not described. Please specify the method or criteria used for this evaluation.
• The diagnostic criteria for NIHL are based on Niskar et al. (2001), a study conducted in a pediatric population. This reference is not appropriate for an adult cohort and undermines the validity of the algorithm. A more suitable reference and criteria should be used instead.
• The rationale for selecting kappa values of 0.4 and 0.6 is unclear. Please specify which scale was used (e.g., Landis and Koch) and justify the chosen thresholds.
• The use of PTA-4 including only up to 3 kHz is problematic. NIHL typically affects higher frequencies, particularly around 6 kHz, which should have been included in the analysis. This limitation needs to be addressed.
• The definition of hearing loss (HL) appears overly simplistic (HTL at one threshold > 25 dB HL). WHO definitions have evolved and are not based on a single threshold. A more nuanced approach should be considered.

Validity of the findings

Discussion and Conclusion
• The manuscript does not convincingly demonstrate that the proposed algorithm is a valid diagnostic tool for ONIHL. While it may have potential as an evaluation or screening tool, its limitations must be more clearly acknowledged. Especially with regards to the beforementioned limitation of the diagnostic complexity of NIHL in general. Therefore, the conclusion appear too bold when considering the limitations of the study methods.
• The reported sensitivity and specificity values (Table 4) as part of the validation process appear unrealistically high, raising concerns about overfitting or oversimplification. The results should be critically examined, and the potential for the algorithm to miss the complexity of ONIHL should be discussed.
• There is inconsistency in terminology: both ONRHD and NIHL are used throughout the manuscript, sometimes interchangeably. This is confusing and should be standardized more. For example, the Discussion mentions both terms, while the Conclusion refers only to ONRHD

Additional comments

Figures:
• The PDF figures are of poor resolution, making them difficult to interpret. Please ensure high-quality images are provided.
• The AUDEXCEL interface is not clearly explained and is difficult to interpret. The x-axis in particular appears problematic, with an unusual gap between 4 kHz, 4.6 kHz, and 8 kHz. This should be corrected or clarified.

---

## Round 0.2 · accepted · Accept

Congratulations on your manuscript being accepted for publication.

With best regards,
Prof. Yoshinori Marunaka, M.D., Ph.D.

·

Basic reporting

The article is clear and unambiguous with the use of professional English throughout. Also, the in-text citations were all relevant and reflected in the reference section.

In addition, the article was well structured: figures, tables were well described and labelled, raw data, the AUDEXEL algorithm, and the user manual of the AUDEXEL were well presented in the supplementary files.

The submission was in order with all the results relevant to the hypotheses

Experimental design

This is an original research article that is within the scope of the journal. Research questions were well defined and were able to bridge the knowledge gap.

The study performed a rigorous investigation with a high technical and ethical standard, and the methods employed were appropriate and detailed, with sufficient information for future studies

Validity of the findings

It is novel. Data on which conclusions are made have been provided. They are robust, statistically sound, and controlled.

Conclusions are well stated and are linked to the research questions.

Additional comments

This is a good study, and it is novel. The authors were able to give recommendations, especially for future studies, based on the limitations of this current study.

Reviewer 3 ·

Basic reporting

-

Experimental design

-

Validity of the findings

-

Additional comments

All the requested revisions have been satisfactorily addressed.